# Optical-Radiation-Calorimeter Refinement by Virtual-Sensitivity Analysis

**DOI:** 10.3390/s19051167

**Published:** 2019-03-07

**Authors:** Lancia Hubley, Jackson Roberts, Juergen Meyer, Alicia Moggré, Steven Marsh

**Affiliations:** 1School of Physical and Chemical Sciences, West Building, Engineering Drive, University of Canterbury Christchurch 8041, New Zealand; lancia.hubley4@gmail.com (L.H.); jackson.roberts@cdhb.health.nz (J.R.); 2Medical Physics and Bioengineering, Christchurch Hospital, 2 Riccarton Avenue, Christchurch 8011, New Zealand; alicia.moggre@cdhb.health.nz; 3Radiation Oncology Department, University of Washington Medical Center (UWMC), 1959 N.E. Pacific Street, Seattle, WA 98195, USA; juergen@uw.edu

**Keywords:** digital holographic interferometry, optical calorimetry, radiation dosimetry, FRED, optical modelling

## Abstract

Digital holographic interferometry (DHI) radiation dosimetry has been proposed as an experimental metrology technique for measuring absorbed radiation doses to water with high spatial resolution via noninvasive optical calorimetry. The process involves digitally recording consecutive interference patterns resulting from variations in the refractive index as a function of the radiation-absorbed dose. Experiments conducted on prototype optical systems revealed the approach to be feasible but strongly dependent on environmental-influence quantities and setup configuration. A virtual dosimeter reflecting the prototype was created in a commercial optical modelling package. A number of virtual phantoms were developed to characterize the performance of the dosimeter under ideal conditions and with simulated disruptions in environmental-influence quantities, such as atmospheric and temperature perturbations as well as mechanical vibrations. Investigations into the error response revealed that slow drifts in atmospheric parameters and heat expansion caused the measured dose to vary between measurements, while atmospheric fluctuations and vibration contributed to system noise, significantly lowering the spatial resolution of the detector system. The impact of these effects was found to be largely mitigated with equalisation of the dosimeter’s reference and object path lengths, and by miniaturising the detector. Equalising path lengths resulted in a reduction of 97.5% and 96.9% in dosimetric error introduced by heat expansion and atmospheric drift, respectively, while miniaturisation of the dosimeter was found to reduce its sensitivity to vibrations and atmospheric turbulence by up to 41.7% and 54.5%, respectively. This work represents a novel approach to optical-detector refinement in which metrics from medical imaging were adapted into software and applied to a a virtual-detector system. This methodology was found to be well-suited for the optimization of a digital holographic interferometer.

## 1. Introduction

Radiation dosimetry is an umbrella term for a family of measurement techniques that quantify the amount of ionising radiation energy absorbed by an object or body. Radiation dosimetry is essential to many branches of science and industry and, in particular, healthcare for cancer treatment. As a radiation dose is defined as the amount of energy absorbed per kilogram of mass, one of the fundamental methods for its measurement is calorimetry, where the temperature change of the absorbing medium is measured and used to calculate the amount of absorbed energy. Conventional dosimetric calorimeters use thermistors embedded in absorption media to measure the temperature increase, but the inclusion of the sensors disturbs the propagation of the radiation field, necessitating the application of cavity-correction factors when calculating the absorbed dose [1]. These calorimeters also tend to be large and rely on stringent environmental control, making them well-suited for primary-standard laboratories, but impractical for clinical environments, where more stable but less direct methods of measuring doses, such as ionisation chambers, are used instead [2].

A proposed alternative to conventional calorimetry is a specialised type of optical interferometry known as digital holographic interferometric (DHI) calorimetry. Optical interferometry describes a family of metrology techniques in which interfering light waves are used to probe any physical parameter of a system that affects the path length of light travelling through it. In general, interferometers rely upon the use of two light fields: one that interacts with the sample of interest, and one that bypasses it. The beams are then superimposed and the resulting interference pattern is analysed to extract information about the parameter under investigation. Holographic interferometry describes an extension of interferometry in which interference patterns of two states are captured and then reinterfered experimentally or digitally in order to extract a two-dimensional map of optical path length difference with a subwavelength resolution. Due to its near-unparalleled sensitivity and noninvasiveness, interferometry is widely used in science and industry [3,4,5].

Holographic interferometry for radiation dosimetry was pioneered in 1971 by Hussman, who used a panchromatic sheet film to capture interferograms of a water cell before and after its irradiation with a high-intensity-pulsed 2 MV electron beam. Isodose contours and percentage depth dose curves were successfully reconstructed from these interferograms [6,7]. The technique was subsequently refined, with Miller et al. employing a photoelectric detector and scanning photodiode in lieu of holographic reconstruction to achieve the fast processing of fringe shifts in interference patterns [8,9,10]. More recently, electronic imaging sensors and computerised reconstruction algorithms have been applied to interferometry, facilitating new investigations into DHI radiation dosimetry [11,12]. In 2014, Cavan and Meyer constructed a prototype DHI radiation dosimeter and successfully determined absorbed-dose distributions in water from a high-dose-rate brachytherapy source, and verified them against values obtained from a treatment planning system (TPS), but this dosimeter was sensitive to environmental fluctuations. Its dose resolution was determined experimentally to be 3.45 Gy, which is greater than the approximately 2 Gy dose delivered in typical radiation-therapy treatment [13]. In 2016, Flores-Martinez et al. expanded on the proof-of-concept prototype, adding improved thermal shielding to minimise the impact of ambient-temperature fluctuations, thereby achieving a dose resolution of 0.3 Gy on doses deposited with a 6 MV clinical photon beam [14]. Significantly, these errors were inside those calculated via TPS.

Recent developments in DHI dosimetry have shown the technique to be capable of generating absorbed dose maps with submillimetre spatial resolution, and have demonstrated its applicability to different types of radiation and diverse field geometries [13,14]. However, the sensitivity of DHI dosimetry to fluctuations in an ambient environment, setup discrepencies, and disturbances to the test cell are barriers to its utilisation in a clinical environment. In addition, the cost of precision optical components makes it prohibitively expensive to experimentally refine prototype dosimeter configurations. As such, computer modelling was identified as a useful tool to test the error response of DHI dosimeters to environmental fluctuations and virtually trial improvements to a prototype design. To this end, Beigzadeh et al. recently reported on their development of a MATLAB model that reliably simulated the fringe patterns produced by a Mach–Zehnder-type DHI calorimeter, reproducing interference patterns obtained in experiments from the existing literature [15].

This paper describes the refinement and sensitivity analysis of a DHI calorimeter via the incorporation of medical-physics quality-assurance techniques into commercial optical modelling software. A virtual dosimeter mirroring the Cavan–Meyer prototype was created in the optical modelling program FRED (Photonics Engineering, Tucson, AZ), using its complex ray-tracing functionality to model interference phenomena. The aim of this research was to show that optical modelling software can be used to accurately simulate a DHI calorimeter, identify its major sources of error, and virtually test improvements to its design. The model, in conjunction with the quality-assurance protocols that were developed, found the major contributor to dosimetric uncertainty to be atmospheric turbulence causing fluctuations in air pressure, temperature, and humidity. The model was used to test improvements to the dosimeter, and it was found that miniaturisation of the dosimeter and equalisation of its path length largely mitigated the impact of environmental fluctuations on the measured dose.

## 2. Materials and Methods

### 2.1. Experimental Dosimeter Prototype

The virtual DHI dosimeter was modelled after the prototype lensless Fourier transform-type (LFT-type) interferometer developed by Cavan and Meyer in 2014 [13]. In the experimental LFT interferometer, a Melles Griot helium–neon (HeNe) laser (IDEX Health and Science, Rochester, NY) with a centre wavelength of 632.8 nm, a spectral bandwidth of 6.25 pm, and a calculated coherence length of 20.4 mm, is used. This laser is passed through a semireflective splitter to create the object and reference beams, as shown in Figure 1. The object beam is expanded into a plane wave and passes through the water-filled test cell, while the reference beam travels through a spherical lens, bypassing the test cell. The reference and object beams are subsequently superimposed, creating an interference pattern which is captured by a monochromatic 1.3 megapixel (1280 × 1024) Pixelink PL-B741 CMOS camera (Navitar Inc., Rochester, NY, USA).

Irradiation of the test cell with ionising radiation creates local variations in water temperature depending on the absorbed dose at each point. This leads to corresponding local variations in water density, thereby altering the refractive index and optical path length traversed by the object beam as it passes through the test cell, changing the interference pattern captured on the CMOS.

The intensity map of the interference pattern recorded on the CMOS is expressed by $I(X_{H},Y_{H})$, where $X_{H}$ and $Y_{H}$ denote the plane of the detector. Holographic reconstruction was carried out via algorithms developed by Cavan and Meyer, which computationally replicate the experimental process of reilluminating the amplitude transmittance map equal to $I(X_{H},Y_{H})$, with the reference beam, denoted $R(X_{H},Y_{H})$. This process is replicated by a modified Fresnel transform, where the reconstructed object wave in the image plane, $O(X_{I},Y_{I})$, is given by
(1)$$O\left(X_{I},Y_{I}\right)=const.\frac{exp(iks)}{i\lambda s}\left[-i\frac{k}{2s}\left({\left(X_{I}\Delta X\right)}^{2}+{\left(Y_{I}\Delta Y\right)}^{2}\right)\right]*F_{\lambda s}^{-1}\left\{I\left(X_{H},Y_{H}\right)\right\},$$
where *k* is the light wavenumber, *s* is the sensor to image distance, λ is the light wavelength, ΔX and ΔY represent the x and y dimensions of the detector pixels, and $F_{\lambda s}^{-1}$ is a discrete inverse fast Fourier transform algorithm scaled by ${\left(\lambda s\right)}^{-1}$. 2D maps of the object wave’s intensity, $I(X_{I},Y_{I})$, and phase, $\Phi (X_{I},Y_{I})$, can then be extracted via the basic complex wave relations
(2)$$I\left(X_{I},Y_{I}\right)={\left|O(X_{I},Y_{I})\right|}^{2},$$
and
(3)$$\Phi \left(X_{I},Y_{I}\right)=\arctan\frac{Im\left[O(X_{I},Y_{I})\right]}{Re\left[O(X_{I},Y_{I})\right]},$$
where $\Phi (X_{I},Y_{I})$ varies between −π and π due to the cyclic nature of the wave phase.

Reconstructed-object wave-phase maps can be extracted from acquired holograms of the test cell in both its unirradiated and irradiated states, and the interference phase between them is calculated by modulo 2π subtraction
(4)$$\Delta \Phi \left(X_{I},Y_{I}\right)=\left\{\begin{array}{cc} \Phi_{1}-\Phi_{2}, & \mathrm{if}\Phi_{1}\geq \Phi_{2}\\ \Phi_{1}-\Phi_{2}+2\pi , & \mathrm{if}\Phi_{1}<\Phi_{2} \end{array}\right..$$

The result, $\Delta \Phi (X_{I},Y_{I})$, is the reconstructed holographic interferogram, which varies between 0 and 2π. An example interferogram is shown in Figure 2, where it can be seen that the three main features are the bright DC term, and the real and twin images of the test cell. The real-image region was manually selected, and a robust 2D phase-unwrapping algorithm was used to convert it from a discontinuous 0 to 2π map to a continuous-phase difference map [16].

The continuous-phase difference map was then converted to a change in optical path length using
(5)$$\Delta OPL\left(X_{I},Y_{I}\right)=\frac{\Delta \Phi (X_{I},Y_{I})\lambda }{2\pi }.$$

The change in refractive index over the width, *d*, of the test cell was then calculated via
(6)$$\Delta n\left(X_{I},Y_{I}\right)=\frac{\Delta OPL(X_{I},Y_{I})}{d}.$$

Temperature change ΔT, which is required to generate this change in refractive index, is found via a third-order polynomial fit to data from Bashkatov and Genina’s study on the temperature dependence of an aqueous refractive index [17]. The deposited dose across the test cell can then be calculated via the calorimetry equation
(7)$$D\left(X_{I},Y_{I}\right)=c_{m}\Delta T\left(X_{I},Y_{I}\right),$$
where $c_{m}$ is the specific heat of water.

### 2.2. Virtual Dosimeter and Sensitivity-Analysis Methods

The commercial optical design software FRED was used to create a virtual model of the Cavan–Meyer prototype dosimeter. Each optical component was separately initialised according to its manufacturer-specified material, geometry, and coating parameters, and the optical elements were integrated into the model using the graphical editing interface in FRED, as shown in Figure 3.

To test the ability of the system to resolve a simple dose distribution, two copies of the water-filled test cell were created. The first was a simple cube of water surrounded by a perspex box, with the refractive index of the water set to correspond to room-temperature ($20{\;}^{\circ }$C) water. The second added a cylinder of warmer water embedded in the room-temperature block, with a refractive index set to correspond to $20{\;}^{\circ }$C water after absorbing a dose of 1 Gy. The refractive indices of these volumes are given in Table 1. These phantoms represented the test cell in its reference and irradiated states, respectively.

Additional test objects, shown in Figure 4, were created to probe the spatial resolution of the system. These test objects reflect phantoms and methodologies typically used in medical imaging [18]. An attenuating slanted edge was created so that the dosimeter’s modulation-transfer function (MTF) could be calculated. The MTF of a system describes its ability to transfer contrast of an image as a function of the image’s spatial frequency. A fencepost phantom was initialised with 20 regions, each with a different number of attenuating and nonattenuating line pairs (lp), with densities from 1 to $20\;{\mathrm{mm}}^{-1}$. Finally, a contrast-detail (CD) phantom was created with a grid of cylindrical dose regions embedded within room-temperature water. These were arranged so that each column corresponded to absorbed doses of 0.01, 0.05, 0.1, 0.5, and 1 Gy, with refractive indices given in Table 1. Similarly, each row of cylinders had diameters of 1, 0.5, 0.25, and 0.1 mm. These virtual test objects were used to analyse the spatial resolution of the modelled dosimeter as per the following protocols.

The slanted-edge phantom was used to give a best-case estimate of the spatial resolution of the system. It was first imaged with the virtual dosimeter, and an interferogram was reconstructed from the generated holograms. An ImageJ (NIH, Bethesda, MD) plugin created by Mitja et al. [19] was used to obtain an edge-spread function (ESF) from the real image region of this interferogram. The ESF was then used to calculate the MTF of the imaging system. By convention, the limiting resolution of a system is taken to be the point where the MTF is equal to 0.1 [18]. It is important to note that calculating the MTF from the interferogram instead of the fully reconstructed interference phase map provided a best-case estimate of the dosimeter’s spatial resolution, discounting the impact of the discrete Fresnel transform and phase-unwrapping steps.

The fencepost phantom was used to estimate the practical-resolution limit of the system. Holograms of the virtual-fencepost phantom were collected and reconstructed, and the resultant image was divided into its 20 constituent spatial-frequency regions. A 2D intensity profile was collected for each region by averaging the intensity of the pixels in each of its columns. These profiles were manually examined, and the practical resolution limit of the system was found to be the spatial frequency at which the line pairs could no longer be reliably counted.

Finally, simulated holograms of the contrast-detail phantom were collected and reconstructed. The reconstructed images were manually examined using a calibrated reference display in order to determine which regions were distinguishable, providing a relative estimate of the contrast and detail resolution of the system depending on the dose and diameter of the visible areas.

### 2.3. Modelling Environmental Uncertainties

The following environmental-influence quantities were identified for modelling and implemented via FRED’s scripting functionality: atmospheric turbulence, atmospheric drift, heat expansion, and mechanical vibration. These were grouped into long-timescale fluctuations (atmospheric drift and heat expansion), which cause homogeneous systematic errors in measured dose and affect the comparability of measurements acquired under different conditions, and short-timescale fluctuations (atmospheric turbulence and mechanical vibration), which cause blurring of the acquired holograms and random defects in measured doses across the test cell.

#### 2.3.1. Atmospheric Drift

The atmospheric refractive index was calculated via the Edlén equations [20], which first calculate the refractivity of standard air for light of wavenumber σ,
(8)$${\left(n-1\right)}_{s}\times 10^{8}=8342.54+\frac{2406147}{130-\sigma^{2}}+\frac{15998}{38.9-\sigma^{2}},$$
then account for air temperature *T* (${}^{\circ }$C) and pressure *P* (Pa),
(9)$${\left(n-1\right)}_{tp}=\frac{P{\left(n-1\right)}_{s}}{96095.43}\times \frac{1+10^{-8}\left(0.601-0.0972T\right)}{1+0.0036610T}.$$

Finally, the impact of atmospheric humidity is factored into the final refractive index,
(10)$$n_{tpf}-n_{tp}=-f\left[3.7345-0.0401\sigma^{2}\right]\times 10^{-10},$$
where *f* is the vapour pressure of water in ambient air (Pa).

To assess the impact of atmospheric drift on dosimetric uncertainty, mean values (μ) and ranges (σ) were calculated for each parameter. The reference atmospheric refractive index was calculated with all the parameters held at the mean value. Next, each parameter was individually changed to its maximum and minimum values in turn, while the others were held at their mean value and the refractive index was recalculated for each parameter combination. Finally, all atmospheric parameters were allowed to vary and a minimum and maximum refractive index value were obtained. Parameter values and the calculated atmospheric refractive indices are recorded in Table 2.

An interferogram of the unirradiated test cell was recorded with the atmospheric refractive index set to its reference value. Interferograms of the test cell with the 1 Gy dose region at the centre were then recorded under each perturbed atmospheric refractive index value. These were each reconstructed against the unirradiated hologram, and the average dose within the dose region was calculated via a 2D averaging algorithm in MATLAB. The defect of each average dose from the actual value (1 Gy) was then calculated for each atmospheric parameter.

#### 2.3.2. Heat Expansion

The average ambient laboratory temperature was determined to be $20{\;}^{\circ }$C, with a range of $\pm 5{\;}^{\circ }$C due to daily variation and weather dependence. These temperature variations slightly alter the length of the beamline through heat expansion, thereby changing the interference phase between reference and object beams, and introducing dosimetric error.

To model heat expansion in the prototype dosimeter, FRED’s scripting interface was used to obtain the spatial coordinates of each optical element. With the origin taken to be the centre of the test cell, each component was then translated according to the following heat-expansion formula:(11)$$\frac{\left(\Delta x,\Delta y,\Delta z\right)}{(x,y,z)=\alpha \Delta T}.$$where $\alpha =23.1\times 10^{-6}\;K^{-1}$ is the heat-expansion coefficient of aluminium, of which the optical breadboard and mounting posts were comprised. In addition, the same formula was used to model the expansion of the test cell, where the thermal-expansion coefficient of perspex $\alpha =69\times 10^{-6}\;K^{-1}$ was substituted for that of aluminium.

A reference interferogram of the unirradiated test cell was collected with no perturbation of the optical elements to represent the $20{\;}^{\circ }$C state. Interferograms of the irradiated test cell were then captured with the optics perturbed to simulate the 15 and $25{\;}^{\circ }$C states, and these were reconstructed against the reference interferogram to determine the degree of error introduced by heat expansion.

#### 2.3.3. Atmospheric Turbulence

Atmospheric turbulence, resulting from fans, air conditioning, and other draughts, is responsible for short-timescale local disruptions in atmospheric temperature, pressure, and humidity, which cause blurring and dosimetric error in the captured interferograms. In order to simulate its effect, the beamline of the interferometer was divided into 9 regions, as shown in Figure 5a.

These were each populated with cubic atmospheric voxels of 2 mm side length as shown in Figure 5b. Each could be individually perturbed with different refractive indices to simulate turbulent atmospheric effects. The mean value (μ) and standard deviation (σ) of each parameter, shown in Table 3, were determined, and a FRED script implementing Gaussian probability model
(12)$$Pr\left(x\right)=\frac{1}{\sqrt{2\pi \sigma }}e^{\frac{{\left(x-\mu \right)}^{2}}{2\sigma^{2}}}$$
was used to assign a random value for each parameter to every voxel in turn. The Edlén equations (Equations (Equation 8)–(Equation 10)) were then used to calculate and assign a refractive index for each voxel. In order to simulate the blurring effects of atmospheric turbulence over the 40 ms acquisition time of the Pixelink CMOS camera, the interferogram recording script was revised so that it collected 40 interference patterns, with the refractive index of the atmospheric voxels perturbed between each acquisition. These holograms were then averaged together in MATLAB to obtain the final image.

Images of the fencepost, slant-edged, and contrast-detail phantoms were captured in order to assess the impact of atmospheric turbulence on the resolution of the dosimeter system.

#### 2.3.4. Mechanical Vibration

The impact of mechanical vibrations on dosimetric uncertainty was also assessed. The amplitude and frequency of displacements induced by common sources (shown in Table 4) were found, and the maximum possible displacement was taken to be the sum of these. A FRED script was created to displace each component by a random length between zero and the maximum displacement along each axis. Each component was then rotated to account for the resultant skew in each coordinate plane.

The time-averaged hologram-acquisition method outlined in Section 2.3.3 was again used here. The period of oscillation was taken to be 1 ms, and 40 holograms were obtained in each set to simulate vibrational blurring over a 40 ms exposure time. Images of the fencepost, slant-edge, and contrast-detail phantoms were captured to assess the impact of vibration on spatial resolution.

### 2.4. Dosimeter Refinements

Two refinements to the dosimeter were tested in order to determine whether they were capable of alleviating uncertainties introduced by environmental fluctuations.

The original dosimeter configuration had very unequal path lengths, with object- and reference-beam lengths of 401.28 and 1235.19 mm, respectively. According to interferometric theory, this contributes to errors caused by homogeneous changes in the atmospheric refractive index or geometric path length, since each optical path is differentially altered according to its length. A new dosimeter configuration was created in FRED with equal length (610.25 mm) reference and object beam paths, as shown in Figure 6a. The uncertainty analyses for atmospheric drift and heat expansion were repeated using this configuration in order to verify that equalising the path lengths mitigated the uncertainties introduced by long-timescale fluctuations.

The second option for dosimetric improvement was miniaturisation, which theoretically mitigates uncertainties introduced by intrabeamline fluctuations by reducing the length of the reference and object paths. A FRED model of a miniaturised dosimeter with equal reference and object path lengths (80–100 mm, depending on the test object being imaged) was created, as per the schematic shown in Figure 6b. The uncertainty analyses for atmospheric turbulence and mechanical vibration were repeated using this configuration in order to verify that the reduced length of the beam paths worked to alleviate blurring effects caused by short-timescale environmental uncertainties.

## 3. Results

### 3.1. Comparison of Simulated and Experimental Results

A sample experimental hologram (Figure 7a) and associated interferogram (Figure 7b) are compared to a simulated hologram (Figure 7c) and associated interferogram (Figure 7d). It can be seen that both interferograms have the same general structure, with a bright central term, a real image, and a twin image. The developed algorithms to reconstruct experimental data were also able to reconstruct the simulated data, indicating that the virtual dosimeter accurately reproduced the output of the experimental configuration. Figure 8 shows the reconstruction of a dose map from the simulated holograms acquired from the 1 Gy absorbed-dose region in the virtual water phantom, demonstrating the model dosimeter’s ability to capture and reconstruct simulated absorbed dose distributions.

### 3.2. Uncertainty Case Studies

#### 3.2.1. Long-Timescale Fluctuations

Reconstructed dose maps of the simulated irradiated phantom were generated for each of the atmospheric refractive indices shown in Table 2, and for the high (25 ${}^{\circ }$C) and low (15 ${}^{\circ }$C) temperature states of the system with simulated heat expansion. These dose distributions were subtracted from the reconstruction of the 1 Gy virtual phantom imaged under reference conditions (shown in Figure 8) to obtain dose defect maps, as shown in Figure 9, for each set of atmospheric parameters and for both heat-expansion cases. The values inside the circular dose region of each map were averaged using MATLAB, and this value was taken to be the mean dose defect generated by the parameter under investigation. The minimum and maximum defects caused by each parameter were then averaged, and this value was used to estimate the percentage of uncertainty contributed by each parameter to the overall absorbed dose. Table 5 summarises these findings.

According to these simulations, the combined drift of all atmospheric parameters generated an average dose defect of $1.5799\times 10^{-4}$ Gy, which was an order of magnitude greater than the defect of $3.0245\times 10^{-5}$ Gy induced by heat expansion. Temperature was found to play the most significant role compared to other atmospheric parameters, with air pressure generating an average defect of $4.0296\times 10^{-6}$ Gy, and humidity causing a $1.4986\times 10^{-8}$ Gy defect.

#### 3.2.2. Short-Timescale Uncertainty Sources

The spatial-resolution characterisation protocols described in Section 2.2 were carried out for the dosimeter with no simulated uncertainties, with simulated atmospheric turbulence, and with simulated mechanical vibration.

The calculated MTF system under each of these uncertainty regimes is displayed in Figure 10, and the best-case limiting resolution where each MTF was equal to 0.1 is given in Table 6. The results of the fencepost image analysis are displayed in Figure 11a–f, where each plot represents an intensity profile across one region of the fencepost phantom. Figure 11a, collected with no simulated environmental uncertainties, exhibited a regular pattern, where each of the 14 fenceposts were clearly distinguishable and countable, indicating that 14 lp/mm is within the limiting resolution of the system. Figure 11b shows a breakdown of this pattern at 15 lp/mm. The fenceposts are no longer distinguishable, indicating that the limiting resolution was exceeded. Figure 11c–f shows intensity profiles of the detector system with simulated atmospheric turbulence, and with simulated mechanical vibration, respectively, with each pair showing the spatial-frequency regions immediately above and below the limiting resolution. The results of this analysis are displayed in Table 6.

As expected, the system with no simulated uncertainty had the best spatial resolution, with a best-case estimate of 34 lp/mm, and a practical estimate of 14 lp/mm. Both the MTF-derived and fencepost-derived limiting resolution estimates indicate that atmospheric turbulence was a more significant source of error than mechanical vibration.

Contrast-detail plots are displayed for each of the uncertainty regimes in Figure 12a–c. Greyed-out regions indicate dose areas that were not distinguishable in the reconstructed image. The contrast-detail results reflect those obtained from the MTF and fencepost analyses, with atmospheric turbulence causing severe reduction in system resolution, and mechanical vibration corresponding to less severe degradation in image quality.

### 3.3. Prototype-Model Refinements

#### Path Length Equalisation

The uncertainty contributed by atmospheric drift and heat expansion were reassessed using a dosimeter configuration with equalised path lengths. Maps of the dose defects generated by each uncertainty simulation are shown in Figure 13, and the estimated uncertainty introduced by each influence quantity are summarised in Table 7, along with the percentage change from the original dosimeter configuration.

The dose defects generated by atmospheric drift and heat expansion on the equalised dosimeter were 96.9% and 97.5% lower than those found using the original dosimeter configuration. This indicates that equalising the lengths of the reference and object beam paths reduces the impact of long-timescale changes in atmospheric refractive index and ambient temperature. This finding is consistent with interferometric theory, which states that the defect should tend toward zero, as the difference in path length is reduced [22].

### 3.4. Miniaturisation

The slanted-edge, fencepost, and contrast-detail phantoms were imaged using the miniaturised system with no environmental perturbations, with simulated atmospheric turbulence, and with simulated mechanical vibrations. Table 8 summarises the spatial resolution of the system found via MTF calculation and the fencepost tests, along with the percentage change from the resolution of the original configuration under each uncertainty simulation. Finally, the contrast-detail plots corresponding to each of the uncertainty simulations are shown in Figure 14, with a red dashed line indicating the visibility boundaries of each of the contrast-detail reconstructions from the original dosimeter configuration.

The miniaturised dosimeter performed better than the original configuration in all spatial-resolution tests. The MTF-derived estimate for the limiting resolution of the miniaturised dosimeter was improved by 29.4% when no environmental perturbations were applied, and by 29.4% and 27.3% for the system subject to atmospheric turbulence and mechanical vibrations, respectively. The fencepost test, which incorporated the effects of phase unwrapping and dose reconstruction, found the spatial resolution with no perturbations to be 28.6% better than the original dosimeter. The resolution of the system under the influence of simulated atmospheric and mechanical perturbations was improved by 58.4% and 41.7%, respectively, by miniaturisation.

Finally, the contrast-detail plots showed the same trend, wherein more dose regions were visible in the reconstructed images of the CD phantom acquired with the miniaturised system compared with those acquired with the larger setup. These results indicate that miniaturisation of the dosimeter yielded a consistent improvement in contrast-detail resolution.

## 4. Discussion and Conclusions

This work showcases an approach to optical-detector refinement and sensitivity analysis that combines methods from medical imaging with the modelling capability of commercial optical design software. This approach was undertaken using a prototype DHI radiation calorimeter as a case study.

A virtual detector was created in Photon Engineering’s optical design program FRED, and a number of simulated phantoms analogous to those used in medical-imaging quality-assurance testing were integrated into the model to characterise its performance and dependence on environmental influence quantities. Variation in atmospheric parameters, ambient temperature, and displacement due to mechanical vibration were simulated through perturbation of the optical model, and the system’s performance was probed under these uncertainty regimes in order to establish the degree and nature of error introduced by each one.

The spatial resolution of the ideal detector was estimated via MTF calculation to be 34 lp/mm, and by imaging a fencepost phantom to be 14 lp/mm. These were reduced to 22 and 7 lp/mm when the effects of mechanical vibrations were simulated, and 17 and 5 lp/mm when atmospheric turbulence was simulated, indicating that atmospheric turbulence had more serious impact upon the spatial resolution of the dosimeter. Long-timescale fluctuations in the atmospheric refractive index and ambient laboratory temperature were found to introduce average dose defects of $1.5799\times 10^{-4}$ Gy and $3.0245\times 10^{-5}$ Gy, respectively, when images of a simulated flat 1 Gy absorbed-dose area were reconstructed.

These virtual quality-assurance tests were repeated on two dosimeter configurations that were proposed as potential improvements to the prototype: a configuration with equalised reference and object path lengths, and a miniaturised configuration. It was found that these improvements alleviated the dosimeter’s dependence on the simulated environmental influence quantities. Equalising the dosimeter’s path lengths reduced the dose defects generated by simulated atmospheric drift and heat expansion by 96.9% and 97.5%, respectively, signifying a large improvement in dosimeter performance. Similarly, miniaturising the dosimeter improved the MTF-estimated spatial resolution by 27.3% and 29.4% for the system undergoing simulated mechanical vibration and atmospheric turbulence, respectively. Similarly, spatial resolution as estimated by fencepost analysis was improved by 54.5% for the system with simulated atmospheric turbulence, and 41.7% for the system undergoing simulated vibrations.

The sensitivity of interferometric systems to small changes in an experimental setup, and the reconstruction steps needed to physically interpret their raw output, make the experimental optimisation of interferometric optical detectors difficult. This simulation-based approach to detector refinement has proven advantageous for analysing detector performance and probing its sensitivity to different environmental influence quantities. Simulating fluctuations in each influence quantity separately allowed the nature and degree of their impact to be assessed in isolation. This provided valuable insight into which improvements would be the most beneficial for refining the prototype and, using QA-inspired virtual phantoms as a surrogate for the endpoint of absorbed dose to water, made it possible to test targeted alterations with the aim of reducing the DHI dosimeter’s dependence on different environmental influence quantities.

The next step in this research is to carry out rigorous experimental verification of the virtual-detector model. Physical phantoms will be developed and incorporated into the interferometer prototype, and identical tests will be carried out experimentally and virtually in order to establish equivalence between modelled and physical results. As the original experiments conducted by Cavan were carried out using an Ir-192 brachytherapy source, it is a priority for future work to reproduce these experiments both physically and via our model dosimeter in order to establish correspondence between the physical and virtual systems’ response to clinical-dose distributions. Additional testing will be carried out using high-energy X-rays generated via synchrotron in order to probe the system’s sensitivity to different radiation-delivery modalities. Once this is achieved, our quality-assurance methodology can be used in conjunction with the model to incrementally refine the detector prototype, with the goal of reducing its uncertainties to clinically acceptable levels.

## Figures and Tables

**Figure 1 sensors-19-01167-f001:**
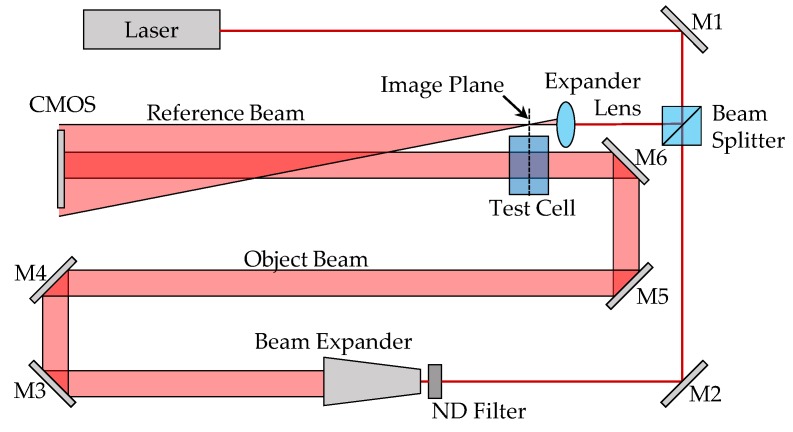
Schematic of the Cavan–Meyer Lensless Fourier Transform Digital Holographic (LFTDH) interferometer. M1–6 are turning mirrors, and the ND filters were a neutral density filter, added to equalise the intensity of the reference and object beams at the CMOS surface.

**Figure 2 sensors-19-01167-f002:**
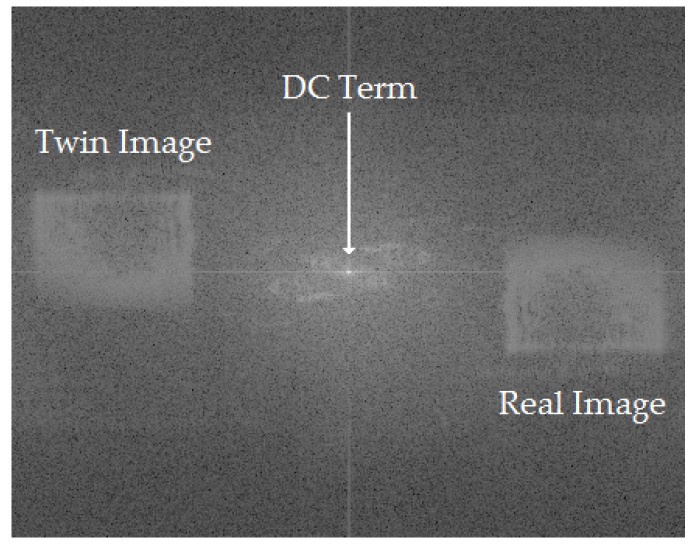
Example of experimentally derived interferogram collected by Cavan [13].

**Figure 3 sensors-19-01167-f003:**
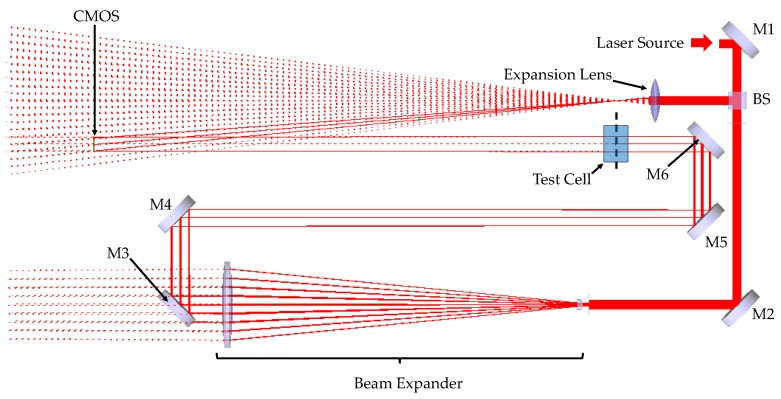
Raytraced FRED model of the LFTDH interferometer.

**Figure 4 sensors-19-01167-f004:**
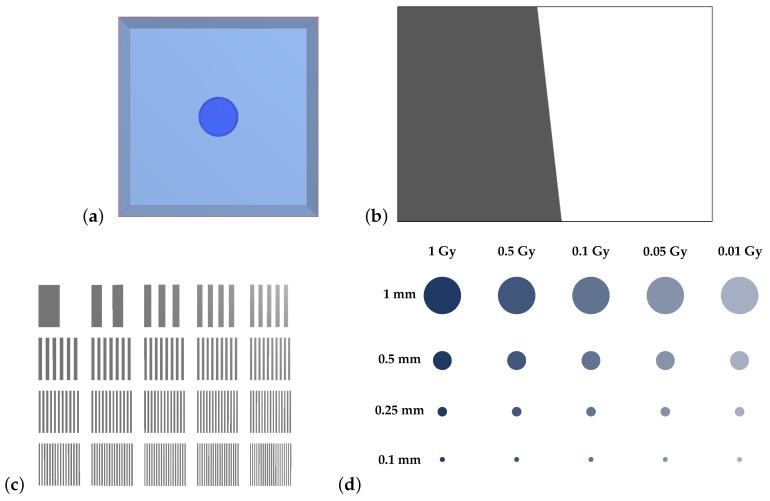
Illustrations of phantom objects created in FRED. (**a**) Flat dose area where the central cylinder represents $20{\;}^{\circ }$C water after absorbing a dose of 1 Gy. (**b**) Attenuating slanted edge for modulation-transfer function (MTF) analysis. (**c**) Fencepost phantom with areas ranging from 1 to 20 lp/mm. (**d**) Contrast-detail phantom with a range of volumes arranged in a grid according to their doses (columns) and diameters (rows).

**Figure 5 sensors-19-01167-f005:**
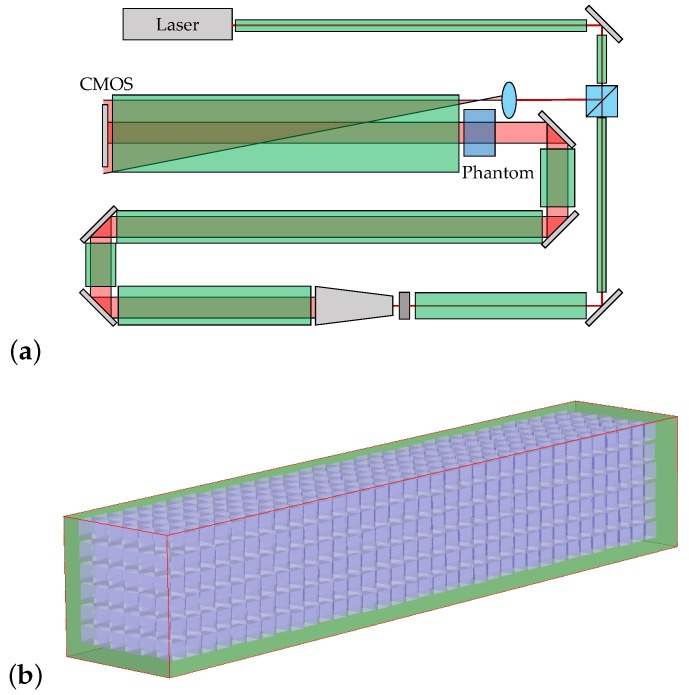
(**a**) Diagram showing the 9 bounding boxes (green) demarcating atmosphere within the dosimeter beamline. (**b**) Screenshot showing one of the boxes populated with atmospheric voxels.

**Figure 6 sensors-19-01167-f006:**
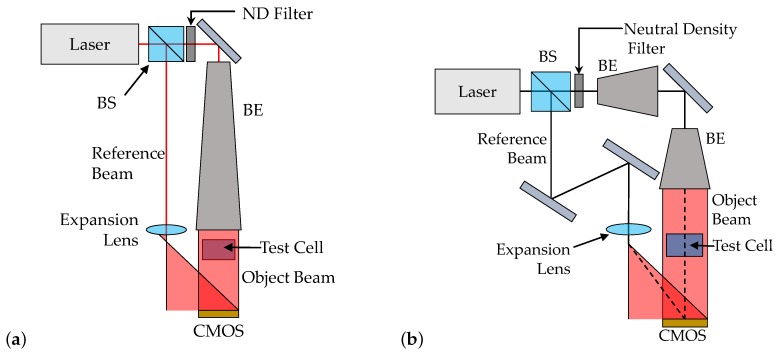
(**a**) Schematic of the FRED model of a dosimeter with equalised beamlines. (**b**) Schematic of the FRED model of the miniaturised dosimeter.

**Figure 7 sensors-19-01167-f007:**
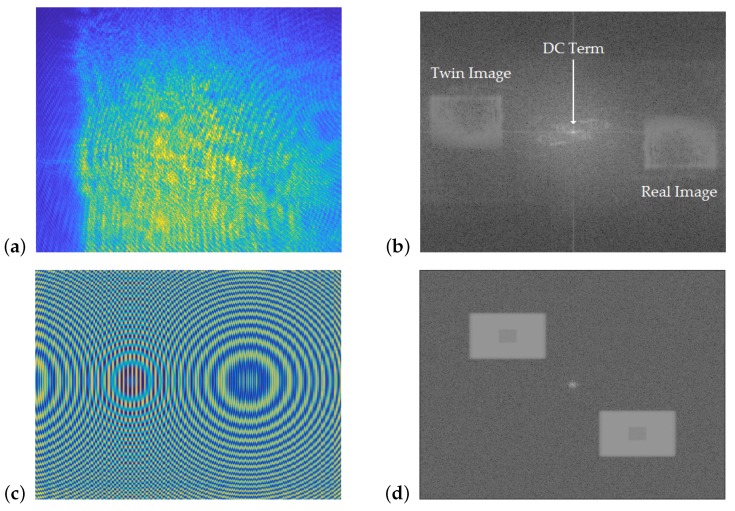
Comparison of experimentally derived and simulated holograms and interferograms. (**a**) Experimentally-derived hologram. (**b**) Interferogram reconstructed from (**a**). (**c**) Simulated hologram. (**d**) Interferogram reconstructed from (**c**).

**Figure 8 sensors-19-01167-f008:**
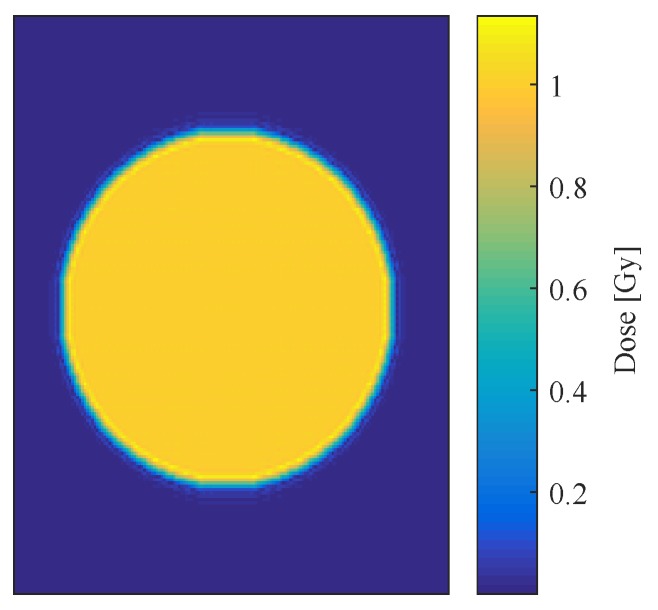
Reconstruction of the dose map for the phantom with embedded 1 Gy absorbed-dose cylinder.

**Figure 9 sensors-19-01167-f009:**
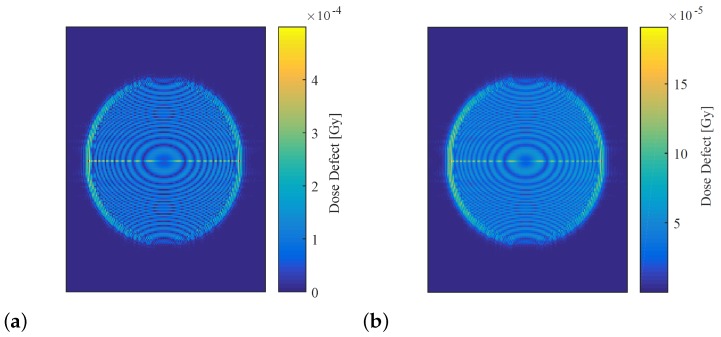
Dose defect maps of the simulated 1 Gy absorbed-dose phantom. (**a**) Dose defect caused by simulated atmospheric perturbation, where every parameter was set to maximise a change in refractive index. (**b**) Dose defect caused by simulated heat expansion of 5 ${}^{\circ }$C. Note the different scales.

**Figure 10 sensors-19-01167-f010:**
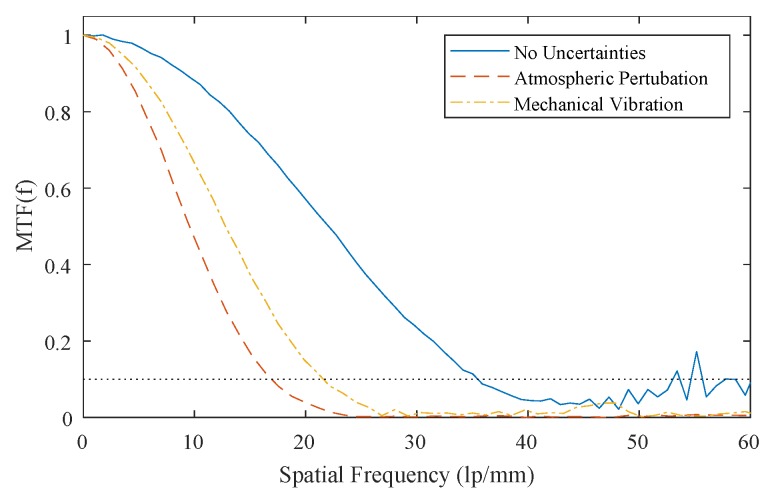
MTF plots of the system with no uncertainties, with simulated atmospheric turbulence, and with simulated mechanical vibrations.

**Figure 11 sensors-19-01167-f011:**
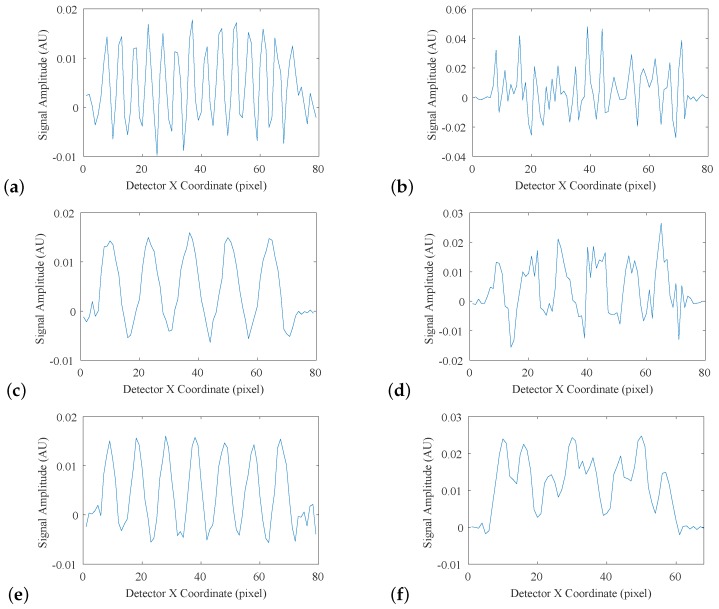
Intensity profiles across different regions of the reconstructed fencepost phantom. (**a**) Collected 14 lp/mm region with no simulated uncertainties; (**b**) collected 15 lp/mm region with no simulated uncertainties; (**c**) collected 5 lp/mm region with simulated atmospheric turbulence; (**d**) collected 6 lp/mm fencepost region with simulated atmospheric turbulence; (**e**) collected 7 lp/mm region with simulated mechanical vibration; (**f**) collected 8 lp/mm fencepost region with simulated mechanical vibration.

**Figure 12 sensors-19-01167-f012:**
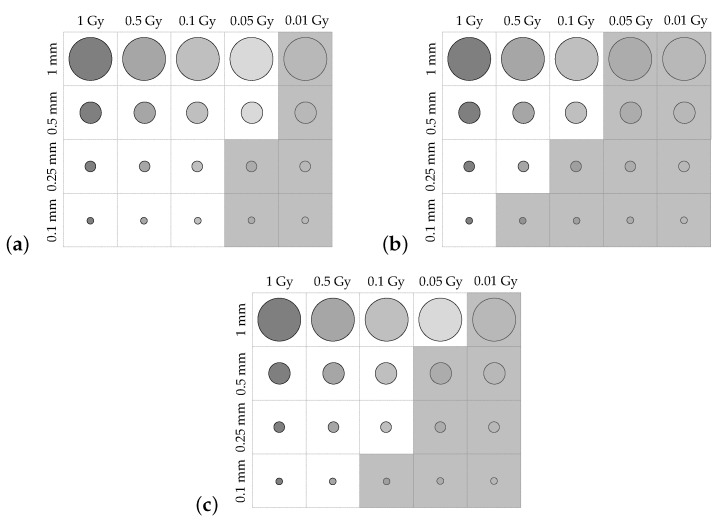
Comparison of contrast-detail plots for the dosimeter with: (**a**) No uncertainties; (**b**) simulated atmospheric turbulence; (**c**) simulated mechanical vibrations. Greyed-out regions indicate voxels that could not be distinguished.

**Figure 13 sensors-19-01167-f013:**
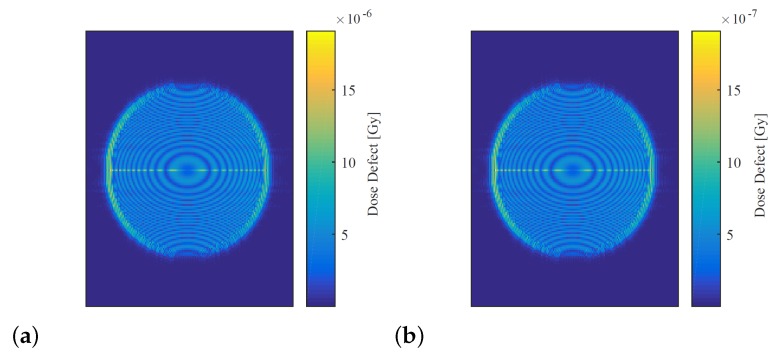
Dose-defect maps generated using the virtual dosimeter with equalised path lengths. (**a**) Dose defect map generated by simulating atmospheric drift. (**b**) Dose defect map generated by simulating heat expansion. Note the different scales.

**Figure 14 sensors-19-01167-f014:**
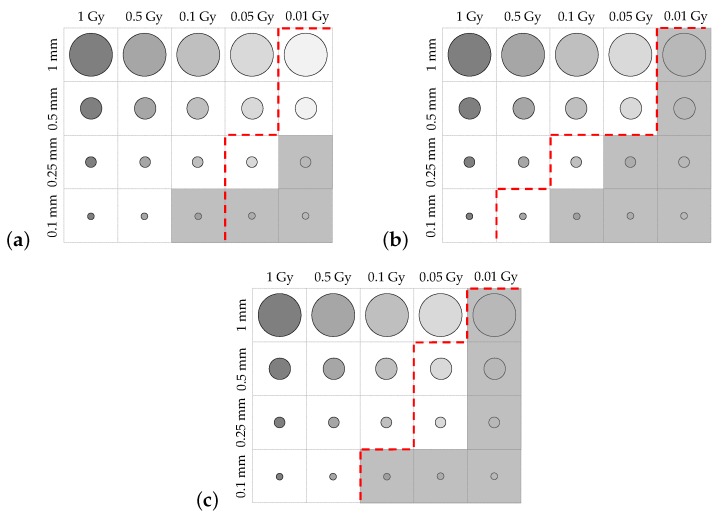
Comparison of contrast-detail plots for the miniaturised dosimeter with: (**a**) no uncertainties; (**b**) simulated atmospheric turbulence; (**c**) simulated mechanical vibrations. Greyed-out regions indicate voxels that could not be distinguished. Red lines indicate the boundary of distinguishable voxels for each uncertainty simulation on the nonminiaturised dosimeter.

**Table 1 sensors-19-01167-t001:** Refractive index of water against different absorbed doses used in the test objects.

Absorbed Dose (Gy)	Water Temperature (${}^{\circ }$C)	Refractive Index
0	20	1.331207965971136
0.01	20.0000024	1.332074521938208
0.05	20.000012	1.332074521112955
0.1	20.000024	1.332074520081388
0.5	20.00012	1.332074511828834
1	20.00024	1.332074501513100

**Table 2 sensors-19-01167-t002:** Atmospheric parameters and their corresponding refractive indices.

Parameter	Parameter Value	Refractive Index
Temperature	$19{\;}^{\circ }$C	1.000272731479275
	$21{\;}^{\circ }$C	1.000270871377762
Pressure	101.315 kPa	1.000271771425499
	101.335 kPa	1.000271825097028
Humidity	48%	1.000271798388954
	52%	1.000271798133571
All	Min	1.000270697986413
	Max	1.000271898261263

**Table 3 sensors-19-01167-t003:** Mean and standard deviation of atmospheric parameters for turbulent air.

Parameter	Mean	Standard Deviation
Temperature	$20{\;}^{\circ }$C	$2{\;}^{\circ }$C
Pressure	101.324 kPa	0.025 kPa
Humidity	50%	2%

**Table 4 sensors-19-01167-t004:** Sources of mechanical vibrations with their amplitudes and frequencies [21].

Source	Amplitude	Frequency
Machinery	$1\times 10^{-2}$ mm	105 Hz
Traffic	$1\times 10^{-1}$ mm	50 Hz
Acoustic	$1\times 10^{-3}$ mm	75 Hz
Building Resonance	1 mm	10 Hz
Building Sway	1 mm	1 Hz
Motors	$1\times 10^{-2}$ mm	275 Hz

**Table 5 sensors-19-01167-t005:** Error generated by simulating atmospheric drift and heat expansion in the virtual dosimeter.

Parameter		Mean Dose Defect (Gy)	Estimated Uncertainty (Gy)
Atmospheric Temperature	Min	$1.4048\times 10^{-4}$	$1.3996\times 10^{-4}$ Gy
	Max	$1.3943\times 10^{-4}$	
Atmospheric Pressure	Min	$4.0306\times 10^{-6}$	$4.0296\times 10^{-6}$ Gy
	Max	$4.0286\times 10^{-6}$	
Atmospheric Humidity	Min	$1.3803\times 10^{-8}$	$1.4986\times 10^{-8}$ Gy
	Max	$1.6169\times 10^{-8}$	
All Atmospheric Params	Min	$1.6549\times 10^{-4}$	$1.5799\times 10^{-4}$ Gy
	Max	$1.5048\times 10^{-4}$	
Heat Expansion	Min	$3.0028\times 10^{-5}$	$3.0245\times 10^{-5}$ Gy
	Max	$3.0462\times 10^{-5}$	

**Table 6 sensors-19-01167-t006:** Spatial-resolution results for the virtual dosimeter under different uncertainty scenarios.

Uncertainty Regime	MTF Resolution	Fencepost Resolution
No Uncertainties	34 lp/mm	14 lp/mm
Atmospheric Turbulence	17 lp/mm	5 lp/mm
Mechanical Vibration	22 lp/mm	7 lp/mm

**Table 7 sensors-19-01167-t007:** Dose-defect uncertainties generated on the equalised dosimeter by atmospheric drift and heat expansion tabulated against the percentage difference from the original configuration.

Uncertainty Source	Estimated Uncertainty (Gy)	Difference
Atmospheric Drift	$4.8632\times 10^{-6}$	−96.9%
Heat Expansion	$7.5003\times 10^{-7}$	−97.5%

**Table 8 sensors-19-01167-t008:** Spatial-resolution results for the miniaturised dosimeter under different uncertainty scenarios tabulated against the percentage difference from the original configuration.

Uncertainty Source	MTF Res.	Difference	Fencepost Res.	Difference
No uncertainties	44 lp/mm	+29.4%	18 lp/mm	+28.6%
Atmospheric turbulence	22 lp/mm	+29.4%	11 lp/mm	+54.5%
Mechanical vibration	28 lp/mm	+27.3%	12 lp/mm	+41.7%

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
