# Peer review of "Optical-Radiation-Calorimeter Refinement by Virtual-Sensitivity Analysis"

_sensors, 2019, doi:10.3390/s19051167_

Round 1
Reviewer 1 Report
Authors demonstrate a DHI radiation dosimetry and study environmental influence to its performance. In general, the article is well written and results are of interests. Still authors should add the following information in the manuscripts:
On page 2, please include the center wavelength and line-width for the used laser, and calculated coherent length;
Figure 3, please mark water cube's location in measurement.
What radioactive source is used in study? Have authors studies system's response when various sources are used?
Author Response
Letter provided in attached PDF.

Reviewer 2 Report
The manuscript describes the refinement and sensitivity analysis of a DHI calorimeter via the incorporation of medical physics quality assurance techniques into commercial optical modelling software. The work is consistent in its scientific results. I believe it is appropriate for your publication.
Author Response
Please find letter in attached PDF.
